# Formation Mechanisms and Kinetic Modeling of Key Aroma Compounds During Qidan Tea Roasting

**DOI:** 10.3390/foods14122125

**Published:** 2025-06-18

**Authors:** Xing Gao, Siyuan Wang, Ying Wang, Huanlu Song

**Affiliations:** Laboratory of Molecular Sensory Science, School of Food and Health, Beijing Technology and Business University, Beijing 100048, China; gaoxing.ing@outlook.com (X.G.); wangsiyuantt@163.com (S.W.); wangying_96118@163.com (Y.W.)

**Keywords:** Qidan tea, roasting time, aroma formation, furan compounds, Carbon Module Labeling (CAMOLA) technique

## Abstract

Understanding the changes in tea aroma and non-volatile substances during roasting is essential for optimizing tea processing and enhancing tea quality. In this study, the Carbon Module Labeling (CAMOLA) technique was employed to simulate the roasting conditions of Qidan, thereby elucidating the formation pathway of the theanine-glucose Maillard system. Combined with sensory evaluation, the results indicated that the floral and fruity aromas of Qidan tea decreased, while the woody, roasted, smoky, and herbal aromas increased with prolonged roasting time. Kinetic modeling demonstrated that higher temperatures favored the production of benzaldehyde, which was directly proportional to the heating temperature. In contrast, pyrazines exhibited zero-order kinetics, influenced by both temperature and time. An increasing trend in furans was observed with rising temperature and extended heating time. The kinetic equations effectively describe the changes in aroma compounds associated with merad, highlighting the differences in the production patterns of aroma compounds under varying roasting conditions. This study provides a theoretical foundation for optimizing roasting parameters to enhance tea quality.

## 1. Introduction

Qidan tea, a highly prized variety of Wuyi Rock Tea, holds a special and esteemed position within the category of Oolong Tea. Renowned for its refined and elegant aroma, a flavor profile that combines a subtle spiciness with a sweet undertone, and a distinct rocky character, Qidan tea is rightfully regarded as one of the treasures of Wuyi Rock Tea [1]. High-quality tea products are the result of a harmonious combination of excellent tea raw materials and sophisticated roasting technology. Different combinations of roasting temperature and time endow Wuyi Rock Tea with its unique and characteristic aroma and flavor [2]. During the roasting process, the flavor of Qidan tea undergoes changes as the temperature rises and the roasting time is prolonged. These changes directly impact the quality and production rate of Wuyi Rock Tea, highlighting the importance of precise control over the roasting process [3].

Pyrazines are a class of compounds that play a crucial and indispensable role in shaping the flavor profile of Oolong Tea. These compounds are responsible for imparting roasted, nutty, and popcorn-like aromas to the tea infusion, which are generated through the Maillard reaction [4]. This research demonstrated that the addition of 5-methyl-2,3-diethylpyrazine, 2-methylpyrazine, and 2,5-dimethyl-3-ethylpyrazine to tea beverages effectively replicated the flavor profile of tea [5]. It has been suggested that 3,5-dimethyl-2-ethylpyrazine may contribute to the aroma of roasted Shuixian [1]. Notably, these specific pyrazines were not detected in studies on the aroma of Tieguanyin, another Oolong Tea from northern Fujian [6], indicating the unique flavor composition of Wuyi Rock Tea.

While pyrazines have received considerable attention in tea aroma research, Strecker aldehydes have been relatively overlooked in previous studies. However, recent research has started to shed light on their significance in tea flavor. For example, Schuh and Schieberle [7] employed molecular sensory science techniques to identify 2/3-methylbutyraldehyde as a key aroma-active compound in Darjeeling black tea. These findings, along with those related to the pyrazine series, suggest that both pyrazines and Strecker aldehydes play crucial roles in the unique aroma profile of Wuyi Rock Tea [8]. Nevertheless, the specific key pyrazine and Strecker aldehyde aroma-active compounds in Wuyi Rock Tea remain largely unclear, and there is a lack of comprehensive research on the effects of different roasting times on the pyrazine components and their formation mechanisms in this type of tea.

The Maillard reaction is a complex and multi-step chemical process that occurs during tea roasting and can be broadly divided into three distinct phases. The initial phase involves the condensation of amino acids with reducing sugars, leading to the formation of Amadori and Heyns products. In the intermediate phase, the sugars and amino acids are cleaved from these products. The final phase is characterized by a series of reactions, including dehydration, fragmentation, cyclization, and polymerization [9]. Wuyi Rock Tea fresh leaves are rich in total sugars, proteins, and free amino acids, which account for 40% to 60% of the total dry weight [10]. This abundant material base, combined with the roasting process, provides favorable conditions for the Maillard reaction to occur. Theanine, which constitutes more than 50% of the total amino acids in tea, has been shown to play a vital role in the production of 2-methylpyrazine, 2,5-dimethylpyrazine, and trimethyl pyrazine [11]. During the production of Wuyi Rock Tea, enzyme catalysis helps to release amino acids and reducing sugars from their bound state, creating a more conducive environment for the Maillard reaction and the formation of pyrazines and Strecker aldehydes during the roasting process. Strecker aldehydes, as important products of the Maillard reaction, can be synthesized through various pathways, such as amino acid-assisted decarboxylation via the α-dicarbonyl group, direct degradation through the Amadori rearrangement, oxidative decarboxylation, and thermal reactions [12].

The field of dynamics, which focuses on the study of rates of change in physical, chemical, or biological systems, offers a powerful tool for understanding the dynamic changes in flavor during the roasting process. Kinetics, a branch of dynamics, has been used to study the rates of change in various systems. Kinetic modeling and gas chromatography-mass spectrometry (GC-MS) can be employed to predict changes in the quality and flavor of kiwifruit during storage [13]. Parker [14] reviewed the development of kinetic models for flavor thermogenesis, especially the kinetic changes during the Maillard reaction, and also mentioned the corresponding kinetic models developed in real food systems that can predict the formation of individual flavor compounds or groups of flavor compounds. It has been demonstrated that 2-furfuryl mercaptan, methyl mercaptan, and 2-ethyl-3-methylpyrazine serve as reliable indicators for predicting the shelf life of roasted coffee products. This prediction is based on the Sulfur/Roast Evolution Index at various storage temperatures, utilizing a first-order kinetic reaction model and the Arrhenius equation to develop predictive modeling results [15].

We have screened the most significant aroma-active compounds produced during the Maillard reaction of Qidan tea, specifically 2/3-methylbutyraldehyde, 2-methylpyrazine, and benzaldehyde, and developed corresponding kinetic equations [16]. Building on this research foundation, this study will: (1) conduct a descriptive sensory assessment of the aroma characteristics of chitin tea at various roasting times; (2) utilize CAMOLA to determine the skeletal origins of these key aroma-active compounds and elucidate the pathways of their production; and (3) establish a corresponding Maillard reaction system using theanine and glucose to verify the accuracy and generalizability of the kinetic equations. The goal of this study is to establish a robust theoretical framework for the roasting process of Qidan tea. This will assist the tea-making industry in reducing its reliance on experiential knowledge, enhancing roasting efficiency, and offering scientific guidance for practical production.

## 2. Materials and Methods

### 2.1. Preparation of Tea Samples

Qidan tea was produced in the Bishiyan area of Wuyishan, Fujian Province, China, located at the geomagnetic coordinates of 117.57° E longitude and 27.42° N latitude. All samples were provided by Qiwei Tea Co., Ltd. (Wuyishan, China) and were roasted in accordance with GB/T 18745-2006 [17], with the roasting temperature maintained at 110 ± 5 °C. The samples were collected from three roasting cages at two-hour intervals. Samples (500 g) were collected from three roasting cages at 2-h intervals. The final samples of Qidan were obtained at 0 h, 2 h, 4 h, 6 h, 8 h, 10 h, and 12 h, respectively. Samples were sealed in aluminum foil pouches and stored in a refrigerator at 4 °C for subsequent analysis.

### 2.2. Chemicals and Reagents

[^13^C_6_]-D-glucose (99%) was purchased from Cambridge Isotope Laboratories (Xenia, OH, USA). Glucose, Theanine, Epigallocatechin gallate were purchased from Nanjing Herbal Source Biotechnology Co. (Beijing, China). D_5_-Theanine was purchased from Mannhag Biotechnology Co. (Shanghai, China). Other reagents include: NaCl (99.5% purity) from Sinopharm Chemical Reagent Company Limited (Shanghai, China); hexane (98% purity) from Fisher Chemical Company Limited (Shanghai, China); and nitrogen (99.999% purity), liquid nitrogen, and ultra-high purity helium (99.9992% purity) from Beijing Asia-Pacific Baifu Gas Industry Co. (Beijing, China) etc.

### 2.3. Determination of Free Amino Acid Content

The 17 free amino acids in the samples were quantitatively analyzed by pre-column derivatization.

Standard preparation: Amino acid mixed standard reagent (1 nmol/μL) was diluted with 0.1 mol/L HCl to form different concentrations of mixed standard into the sample and quantified by external standard method. Theanine standard: 0.01 g of theanine standard was diluted to 10 mL with 0.1 mol/L HCl and prepared as 1 mg/mL solution. Sample preparation: 0.4 g of tea powder + 20 mL of 0.1 mol/L HCl solution (20 mg/mL), centrifuged at 10,000 r/min for 10 min, the supernatant was taken and passed through 0.22 μm polyethersulfone (PES) membrane. Amino acid determination: A Zorbax Eclipse-AAA (4.6 mm × 150 mm, 5 μm) column was used for the separation of each amino acid. The mobile phases were as follows: Salt phase A: 40 mmol-L-1 NaH_2_PO_4_ aqueous solution (adjusted to pH = 7.8 with NaOH); Organic phase B: methanol-acetonitrile-water solution (45:45:10, V/V/V). The mobile phase flow rate was set at 1 mL/min, the injection volume was 1 μL, the UV detection wavelength was 338 nm, and the column temperature was set at 40 °C. Theanine was quantified separately from other amino acids due to the difference in concentration range [5].

### 2.4. Descriptive Sensory Evaluation Methods

The overall flavor profile of Qidan tea was assessed before analyzing the flavor components of pre- and post-roasted samples. The sensory evaluation team consisted of 12 members (female: 7, male: 5) with an average age of 27 years. In order to minimize the influence of other external factors on the results of the sensory assessment and to ensure the independence and accuracy of the results, we ensured that the environment was tranquil and free of odors when the sensory assessment was conducted. The assessments were done independently and there was no interaction during the process. The interval between sensory assessments was 30 s between samples, which helped to restore the assessor’s olfactory perception. The scoring system of the study was based on a 5-point scale, where 1 is an interval scale and there are 6 different scales in total: 0 for unrecognizable, 1 for just recognizable, 2 for weak perception, 3 for moderately strong perception, 4 for strong perception, and 5 for very strong perception. The study conducted a blind sensory assessment of the samples to be tested using a random number which was composed of three random digits [18]. The flavor descriptor system of Wuyi Rock Tea covers floral (the fresh aroma of roses), fruity (the aroma of fruits), woody (the aroma of wood), roasted (the smell of popcorn, the aroma of fruits roasted hard), herbaceous (the fresh aroma of grass), smoky (the smell of burnt, smoky flavor), medicinal herbaceous (the aroma of medicinal herbs), and metallic (the smell of metal products), for a total of 8 flavor descriptors. Whether the flavor could be perceived as a whole was indicated by whether it was preferred or not. Three replicate assessments were required for each sample. The final data were derived based on the mean values.

The experimental steps for sensory evaluation are as follows: To prepare the tea for evaluation, begin by preheating the evaluation cups with boiling water. Accurately weigh 5.0 g of tea leaves. Next, quickly add 250 mL of boiling water, maintaining a tea-to-water ratio of 1:50. Cover the container and steep the tea for 5 min. After steeping, promptly transfer 10 mL of the tea infusion into a 40 mL odorless vacuum bottle. To minimize the evaporation of the tea’s flavor, quickly secure the bottle with its cap and place it in a water bath set to 55 °C to facilitate the release of the flavor.

### 2.5. Model Reactions

Two model reaction systems, System A: Theanine (500 μmoL), Glucose (500 μmoL) and two blank control systems, System B: Theanine (500 μmoL), System C: Glucose (500 μmoL). A total of 500 μmoL of each reaction was mixed with ultrapure water to 10 mL and the pH was adjusted with sodium hydroxide to 5.5. After mixing, the reaction was placed in a pressure-resistant reaction flask and reacted in a metal bath at 110 °C for 5 h. After the reaction, the mixture was immediately removed from the metal bath, cooled in an ice bath, and then refrigerated at 4 °C for 24 h pending subsequent analysis.

### 2.6. Solid-Phase Microextraction (SPME)

Before the formal analysis, tea infusions were prepared for each tea sample following China’s national standard “Sensory Evaluation Methods for Tea” (GB/T 23776-2018 [19]). The process was as follows: Weigh 2 g of tea samples and brew them with 100 mL of boiling water (tea-to-water ratio: 1:50), then cover and steep for 3 min. Next, weigh 5 g of the brewed tea broth and transfer it into a 30 mL headspace vial. Add 1.5 g of NaCl and 1 μL of 2-methyl-3-heptanone (0.816 μg/μL, dissolved in n-hexane) to the vial. Incubate the vial in a thermostatic water bath (Shanghai Ge Trading Co., Ltd., Shanghai, China) at 55 °C for 20 min. Subsequently, perform SPME extraction using a 2 cm divinylbenzene/carboxen/polydimethylsiloxane (DVB/CAR/PDMS) extraction fiber (50/30 µm, Supelco, Bellefonte, PA, USA) at 55 °C for 40 min. After extraction, the fiber was inserted into the inlet of the gas chromatography instrument and desorbed at 230 °C for 5 min [20].

### 2.7. GC×GC–MS Analysis

This study utilized GC×GC-O-MS instrumentation, comprising an Agilent 8890 GC coupled with a 5977B MS. It was equipped with a primary polar DB-WAX capillary column (30 m × 0.25 mm, 0.25 µm film thickness; Agilent Technologies, Beijing, China) and a secondary medium-polarity DB-17 MS column (1.85 m × 0.18 mm, 0.18 µm film thickness; Agilent Technologies). The analytical method was adapted from that described by Yang [5], with appropriate modifications. The initial temperature was set at 40 °C for 3 min, followed by an increase to 230 °C at a rate of 4 °C/min, maintained for 5 min. The inlet temperature was established at 230 °C, and helium was used as the carrier gas at a flow rate of 1 mL per minute.

The inlet temperature was set to 230 °C, and the carrier gas used was helium, flowing at a rate of 1 mL/min. The mass spectrometry conditions were as follows: 230 °C for the ion source, 280 °C for the transmission line, and 150 °C for the quadrupole. The electron bombardment ionization mode was employed, with a scanning range of m/z 29 to 500 and an electron energy of 70 eV. Each sample was analyzed three times.

### 2.8. Carbon Module Labeling (CAMOLA) Experiment

The isomorphic thermal reactions involved mixing ^13^C_6_-labeled glucose with unlabeled ^12^C_6_-glucose in a 1:1 ratio. Additionally, the thermal reaction of D_5_-labeled theanine with unlabeled glucose in a 1:1 ratio was conducted to infer the attribution of the carbon atoms of the volatile compounds in the system. The volatile compound proportions of isotopomers were calculated by normalization of the peak areas of the selected ions from M^+^ to M^+^ + n, where M^+^ is the molecular ion, and n is the number of labeled carbons in an isotopomer. Before normalization of peak areas of the selected ions, the isotopomer proportions were corrected by subtracting the naturally occurring percentages of ^13^C (1.10%), ^33^S (0.76%), and ^34^S (4.20%). The loss of hydrogen frequently observed with the molecular ion in EI-MS was also corrected in the labeled molecular ions by the ratio (M^+^ − 1)/M^+^ [21].

### 2.9. Kinetic Studies Analysis

With the extension of roasting time, the concentrations of the main differential aroma-active compounds in tea changed to different degrees. In order to investigate the formation pattern of these characteristic compounds, the kinetics of these substances were further examined using the experimental methods of Wang and Yu [16,22]. The kinetic equations are as follows:(1)$$\frac{dA}{dt}=K{*A}^{n}$$

The letters *A*, *t, K*, and *n* in the formula represent the content of volatile compounds, the reaction time, the reaction rate constant, and the reaction energy level, respectively. By integrating kinetic Equation (1), we obtained the following linear Equation (2):(2)$$A=A_{0}+K_{t}$$

A_0_ represents the initial concentration.

The first-order kinetic equation is as follows:(3)$$\ln A=\ln A_{0}+K_{t}$$

The fit of the regression curve between the measured values and model calculations is represented by R^2^, where the closer the value of R^2^ is to 1, the closer the model is to the measured values.

### 2.10. Statistical Analysis

All the results are expressed as the mean ± standard deviation (average ± SD). The mean and ANOVA of all the aromatic compounds were collated and analyzed using Excel 2019 and SPSS Statistics 27 software. Statistical significance was determined at *p* ≤ 0.05.

## 3. Results and Discussion

### 3.1. Analysis of Descriptive Sensory Evaluation Results of Different Roasting Time of Qidan

Based on the sensory scores of the seven samples shown in Figure 1, it was found that the aroma patterns of the Qidan samples varied at different roasting durations. The scores of aroma profiles of floral, fruity, grassy, and metallic aroma gradually decreased with the increase of roasting time, which was replaced by the scores of aroma profiles of woody, roasted, smoky, and herbal aroma gradually increased. At the beginning of roasting, the overall aroma profile can be summarized as fruity, floral and fresh, and after roasting, the aroma profile changes from fruity, floral and fresh to nutty, roasted and smoky. Throughout the roasting process, the change in aroma showed a clear trend, which may reflect the effect of roasting on the aroma profile of the Qidan samples. This trend change reflects the dynamic evolution of chemical reactions during different roasting stages, and the chemical composition in the Qidan samples may change with increasing roasting time, leading to the evolution of the aroma profile. For example, with increasing roasting time, changes in thermodynamic and kinetic parameters may occur, which may cause the production or consumption of different volatile compounds, thus altering the aroma profile of the Qidan samples. Therefore, the study of the variation of aroma profiles of Qidan samples with roasting time can help to understand the chemical reactions that occur during the roasting process and the mechanisms of aroma formation.

### 3.2. Determination and Analysis of Amino Acid Content

The content of 17 free amino acids was determined, and the standard curve and results are shown in Table 1. From the table, the highest content of free amino acids in Qidan is theanine. Theanine is the characteristic amino acid of tea and is an amide compound with 40% of the free amino acid content, which has a very strong influence on the quality of tea and influences the formation of tea flavor along with other amino acids. Similarly, theanine is crucial in determining the taste of tea broth, theanine has a caramelized aroma and a fresh, crisp flavor similar to monosodium glutamate (MSG), which relieves bitterness and increases the freshness and sweetness of the tea broth. The higher its content, the weaker the astringency of the tea [23]. Perhaps theanine is not involved in the process of protein production, so the content of theanine in tea is higher compared to other amino acids. The content of theanine decreases with increasing roasting time, which may be due to the degradation of theanine with increasing roasting time or due to its involvement in the Maillard reaction, which produces more abundant aroma products. Since theanine is the ethylamine inducer of glutamic acid and the two can be interconverted, this is one of the reasons for the higher glutamic acid content in Qidan. The content of glutamic acid likewise decreases with roasting time. In contrast, the content of the other amino acids was very low in the samples, and although there was a tendency for their content to decrease with roasting time, their content was too low to have a high impact on the aroma. Isoleucine and leucine correspond to 2-methylbutyraldehyde, 3-methylbutyraldehyde, respectively. These are aroma compounds that are Strecker degradation products, for the Maillard reaction related products, and also the focus of the research in this chapter. However, the levels of these two amino acids were too low, and leucine could not even be detected in samples roasted for 4 h. Despite this, the concentrations of both 2-methylbutanal and 3-methylbutanal were relatively high, so we can infer that 2-methylbutanal and 3-methylbutanal may also be formed at very low amino acid concentrations. In the course of studies on the flavor properties of green tea, it was observed that leucine and isoleucine were identified in various green teas, and in view of the relative simplicity of green tea processing of fresh tea leaves [24], it was hypothesized that isoleucine and leucine might be involved in the Maillard reaction mainly in the drying of Wuyi Rock Tea’s gross tea or in the processing of gross tea.

### 3.3. Labeling Glucose to Infer the Source of Aroma Compounds

The study of the metabolic pathways of glucose and theanine precursors by stable isotope labeling technique revealed the carbon source specificity and biosynthetic mechanisms of different aroma compounds (Table 2 and Table 3). Analysis of the isotopic distribution patterns of various aroma compounds showed that there were significant differences in their precursors.

Among the aldehydes, 3-methylbutyraldehyde and benzaldehyde both showed 100% unlabeled [M] in the fully labeled/half-labeled system, suggesting that their carbon skeletons may be entirely derived from unlabeled sources or that there are specific metabolic bypasses, implying that these two aldehydes may not be involved in the conventional metabolic pathways of labeled precursors. In contrast, hexanal was close to a 1:1 ratio of [M + 3]:[M + 5] in the fully labeled system and close to a [M]:[M + 3]:[M + 5] = 1:1:1 distribution in the half-labeled system, confirming that it can be generated via the Maillard reaction involving glutamic acid/theanine and contains at least one carbon atom of amino acid origin.

Furfuryl compounds exhibit a complex carbon source profile. Furfuryl alcohol did not detect [M] in the fully labeled system, and the half-labeled system presented [M]:[M + 5] = 1:1, which, combined with its fully labeled [M + 4]:[M + 5] = 2:3, suggested the existence of a dual pathway of glucose metabolism and amino acid alcoholization. The distribution pattern of furfural with the predominance of the fully labeled system [M + 5] (100%) and the half-labeled system [M]:[M + 4]:[M + 5] ≈ 1:1:1 confirms that it can be formed through multiple pathways such as dehydration of pentose sugar, decomposition of hydroxymethylfurfural, and cyclization of Strecker’s degradation products, which is consistent with the dual pathway mechanism of generation proposed by Zhang et al. [25].

The isotopic distribution pattern of pyrazines reveals their unique carbon and nitrogen source mechanisms. The basic pyrazine structure presents 100% [M + 4] in the fully labeled system and [M]:[M + 2]:[M + 4] = 2:5:3 in the half-labeled system, which, combined with the structural features of its nitrogen-containing aromatic ring, suggests that the carbon skeleton originates exclusively from the C-2 position of glucose while the nitrogen atom is derived from the amino acid [26]. Notably, the pyrazine derivatives with different substituents show specific cleavage patterns: the cleavage feature at the C-2/3 position of 2-methylpyrazine, the preferential cleavage pattern at the C-3 position of 2,5/2,3-dimethylpyrazine, and the contribution feature at the C-5/6/7 position of 2-vinylpyrazine, which together construct a precise localization profile of the glucose carbon atoms in the pyrazine ring.

The isotopic distribution of pyrroles further confirms the diverse transformations of sugar metabolites. 2-Pyrrolecarboxaldehyde’s characteristic [M + 4]:[M + 5] ≈ 1:1 in the fully labeled system, combined with the complex distribution of [M]:[M + 1]:[M + 2]:[M + 3]:[M + 4] ≈ 2:2:1:3:1 in the half-labeled system, suggests that it may be generated through a cyclization reaction involving the C-3 position of glucose. In contrast, the 100% fully labeled [M + 7] versus half-labeled [M + 5]:[M + 7] ≈ 2:1 characterization of 2-acetylpyrrole reveals that its acetyl group may originate from a specific cleavage product of the glucose C-5 site.

Particularly noteworthy is the 100% fully labeled [M + 4] versus half-labeled complex distribution pattern ([M]:[M + 1]:[M + 2]:[M + 3]:[M + 4] ≈ 3:1:1:1:3) of 2,3-butanedione, an isotopic distribution feature that not only confirms that its carbon source is exclusively from glucose, but also suggests the existence of two distinct metabolic pathways: the primary pathway is a single glucose molecule degradation and the secondary pathway may involve the interaction between two glucose molecules. This dual-path metabolic mechanism provides important clues for understanding the complex transformation of sugar precursors.

The results show that the formation of aroma compounds in the Maillard system is characterized by a remarkable diversity of metabolic pathways, including glucose alone (e.g., 2,3-butanedione), sugar-amino acid interactions (e.g., hexanal), and multi-pathway transformations of precursors (e.g., furfural).

Isotopic tracer studies of thermally reactive systems of deuterium-labeled theanine, combined with the experimental data in Table 2 and Table 3, reveal the specific contribution of theanine in the formation of aroma compounds and its metabolic transformation mechanism. In a previous study, Guo [27] produced amides by heating theanine alone and compared them with the aroma-active compounds detected, highlighting that theanine may be a precursor for the volatiles that contribute to the aroma of tea during preparation. Among furans and aldehydes, furfuryl alcohol, hexanal, and furfural all showed 100% [M] in the fully labeled system, a phenomenon that suggests that the main carbon skeleton of these compounds originates from unlabeled glucose metabolites. It is noteworthy that theanine may be involved in its generation through two indirect pathways: first, theanine undergoes Strecker degradation in a thermal reaction to produce ethylamine and glutamic acid, and these degradation products form the target compounds with glucose derivatives via Maillard reaction; second, glutamic acid is directly involved in the synthesis of hexanal as an amino acid precursor.

The isotopic distribution pattern of the pyrrole analogs reveals a more refined mechanism of precursor assignment. 1-Ethyl-1H-pyrrole exhibits a 100% [M + 5] signature in both the half-labeled and fully labeled systems, and this unique isotopic distribution suggests that the molecular structure has a clear dichotomous origin: the carbon skeleton of the pyrrole ring is derived exclusively from glucose metabolites (corresponding to the unlabeled carbon source). In a previous study, Yu [28] further demonstrated that pyrroles can be produced by constructing a Maillard reaction model system for glucose and lysine. The nitrogen atom and some carbon atoms in the ethyl substituent, on the other hand, are derived from the ethylamine component of theanine.

This finding confirms that theanine can be involved in the construction of heterocyclic compounds through the directional transfer of the N-ethylamine group in a thermally reactive system, which is manifested in the specific condensation reaction of the ethylamine component with the five-membered ring structure derived from glucose.

The combined experimental results show that theanine exhibits a dual mode of action in food thermal processing systems: it can participate in the Maillard reaction as an intact amino acid (e.g., the glutamic acid component is involved in the synthesis of hexanal), and it can also generate an active fragment through intramolecular cleavage (e.g., the ethylamine group is involved in the construction of pyrrole analogs) [5]. This structural dissociation-reorganization mechanism provides new perspectives for resolving the molecular contribution of theanine in flavor formation, especially the directional transfer properties of its ethylamine component.

### 3.4. Establishment of Maillard Reaction Model and Kinetic Equation

Taking theanine, glutamic acid, and glucose as the research objects, the accuracy and universality of the kinetic equations were verified, and the corresponding Maillard reaction system was established to study the interactions between these substances and the generation of aroma substances under different roasting conditions. Based on the experimental results, the most representative substances were selected to fit the kinetic equations.

As can be seen from Figure 2, benzaldehyde, as a representative aldehyde, shows a significant correlation between its content change and roasting conditions. With the increase of heating time, the amount of benzaldehyde increased continuously, indicating that longer heating time is favorable to the production of this substance; at the same time, the increase of heating temperature will directly promote the increase of benzaldehyde content, reflecting the accelerating effect of temperature on the reaction process. From the fitting results of the kinetic model, the generation of benzaldehyde at 100 °C conformed to the zero-order kinetic equation, which indicated that the reaction rate at this temperature was independent of the concentration of the reactants, and might be limited by other rate-control steps; whereas, the first-order kinetic equations were followed at 110 °C and 120 °C, which indicated that the reaction rate and the concentration of the reactants were linearly correlated, and that high temperature environment significantly enhanced the reaction activity. In a word, the high temperature conditions were more favorable for the formation of benzaldehyde, and the heating temperature was positively correlated with the rate of formation. The rate of benzaldehyde formation was sluggish at low temperatures but was greatly accelerated at high temperatures.

In this study, the synergistic mechanism of temperature-time parameters on the formation of the products of the Melad reaction was revealed by systematically monitoring the generation pattern of typical pyrazines under different heat treatment conditions. As shown in Figure 2 and Table 4, the generation of 2-methylpyrazine showed typical temperature-dependent characteristics. The compound was not detected in the initial stage (0 h) and in the low temperature treatment group (100 °C and 110 °C treatment for 2 h). When the temperature reached 120 °C, 2-methylpyrazine began to be formed, and its kinetic curve showed a slow accumulation period in the first 4 h, followed by a rapid generation phase, and finally reached the peak concentration at 12 h. The reaction was carried out at a low temperature (0.5 °C). It is noteworthy that, although the reaction process showed a staged rate change, the overall conformity to the zero-order kinetic model may be related to the continuous supply of precursor substances in the reaction system.

For the more structurally complex 2,6-dimethylpyrazine (Figure 2, Table 4), higher activation energy conditions are required for its generation. In the range of 100–110 °C, the concentration of the compounds remained below the detection limit even after extending the treatment time to 12 h. The concentration of the compounds in the range of 100–110 °C was lower than the detection limit. When the temperature was raised to 120 °C, only trace presence was detected for the first 2 h, but the concentration increased significantly after the treatment time exceeded 4 h. The concentration of the compounds in the range of 100–110 °C was significantly higher than the detection limit. Despite the significant nonlinear characteristics of the generation process, the data could still be well fitted by a zero-level kinetic equation, which is in agreement with the findings of Rojas et al. [29] in a cocoa bean roasting system.

The above results suggest that the generation of pyrazines is not only controlled by the temperature threshold, but also requires sufficient thermal interaction time to complete the conversion of precursor substances and molecular rearrangement process. Higher temperatures (≥120 °C) can effectively reduce the activation energy of the reaction and induce the formation of key intermediates from aminocarbonyl compounds via the Strecker degradation pathway, which can then be used to generate pyrazine derivatives with different substituents via free radical coupling reaction [30].

The study of the mechanism of furan production based on the kinetic modeling of the Melad reaction revealed the regulation of 2-methylfuran formation by roasting conditions (Figure 2, Table 4). The experimental data showed that the presence of 2-methylfuran was not detected at the initial stage of the reaction (0–4 h), a latent period characterized by a close correlation with the process of accumulation of the intermediates of the Maillar reaction. When the thermal processing time exceeded 4 h, the compound began to form significantly and showed a time-dependent accumulation trend at all three temperature gradients of 100 °C, 110 °C, and 120 °C, confirming that it is a typical thermally induced secondary product.

The results of the kinetic model fitting showed that the temperature had a significant regulatory effect on the reaction level: under the condition of 100 °C, the generation of 2-methylfuran conformed to the zero-level kinetic characteristics, which indicated that the reaction rate was mainly controlled by the thermal decomposition energy barriers at this temperature; when the temperature was raised to 120 °C, the system showed an obvious accelerated generation phenomenon after 8 h. At this time, the kinetic process was more consistent with the first-level reaction model, which implied that the concentration of intermediate or products had become the rate-limiting factor.

Further analysis revealed that there was a significant temperature-time synergistic effect in the reaction system: the final concentration of 2-methylfuran increased with every 10 °C increase in temperature for the same thermal processing time, while the extension of the time to 12 h at a constant temperature resulted in the increase of the high concentration of the product. This nonlinear growth characteristic is directly related to the cascade effect of the successive reactions of reducing sugar degradation and amino acid decarboxylation in the Maillard reaction, indicating that the generation of furans is a combined result of the multi-step transformation of multiple precursor substances. The exponential kinetic equations developed in the study provided a quantitative model for predicting the furanoid production under different roasting parameters.

## 4. Conclusions

In this investigation, glucose and theanine were employed to simulate the Maillard reaction system, aiming to explore the carbon-skeleton origins of crucial aroma-active compounds under various roasting conditions. Through a detailed analysis of the aroma compound generation patterns, it was observed that the product release profiles of the Maillard reaction exhibited variations across different roasting scenarios, with the generation rate being influenced by both temperature and duration. The study successfully established the correlation between the generation patterns of specific compounds, namely benzaldehyde, 2-methylpyrazine, 2,6-dimethylpyrazine, and 2-methylfuran, and the roasting conditions employed. Moreover, the temporal dynamics of flavor compound content were accurately described using kinetic equations. This research not only underscores the pivotal role of the Maillard reaction in the roasting process of Qidan tea but also offers valuable insights and practical guidance for optimizing production processes.

## Figures and Tables

**Figure 1 foods-14-02125-f001:**
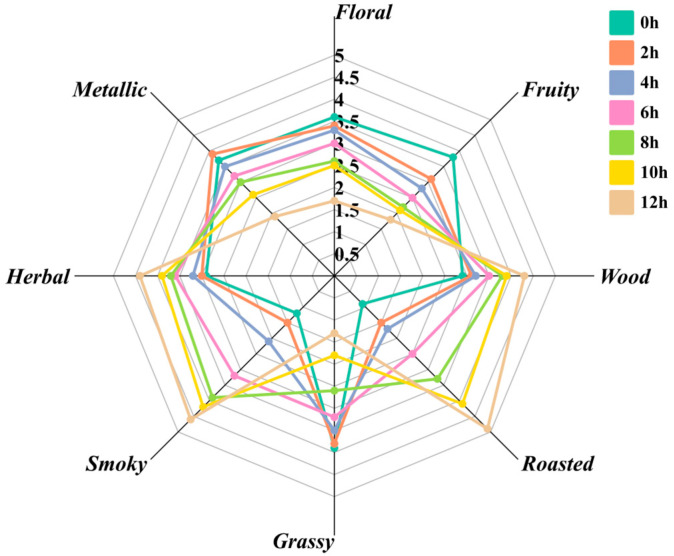
Descriptive sensory evaluation contour plot of Qidan samples with different roasting time.

**Figure 2 foods-14-02125-f002:**
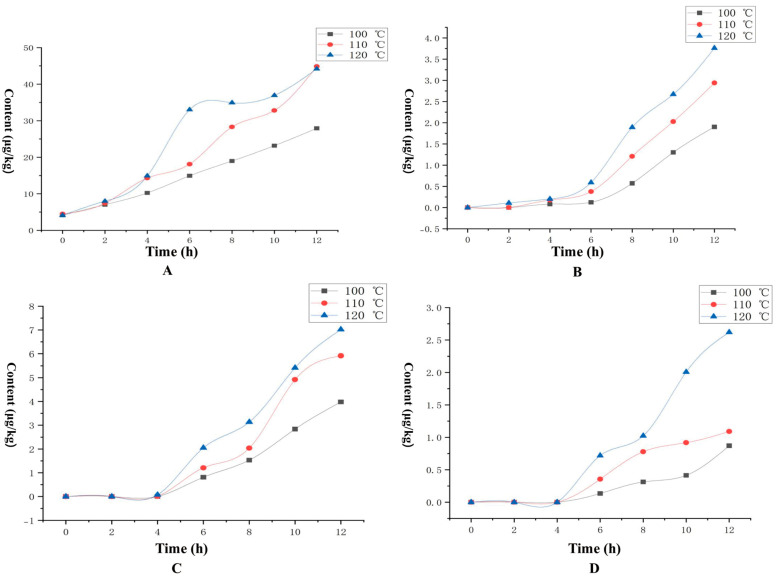
Changes in concentration of major aroma-active compounds ((**A**)−benzaldehyde, (**B**)−2-methylpyrazine, (**C**)−2,6-dimethylpyrazine, (**D**)−2-methylfuran) at different combinations of temperature and time.

**Table 1 foods-14-02125-t001:** Amino acid content in Qidan tea under different roasting times.

No	Amino Acids	Molecular Weight	Standard Curve	R^2^	Concentration (mg/mL)						
					0 h	2 h	4 h	6 h	8 h	10 h	12 h
1	Asp	133.1	y = 106.13x + 2.263	0.9926	28.35 ± 3.14 ^a^	29.11 ± 4.21 ^a^	23.19 ± 3.14 ^b^	20.42 ± 2.41 ^b^	16.27 ± 0.14 ^c^	18.13 ± 3.14 ^c^	23.43 ± 6.27 ^b^
2	Glu	147.1	y =186.24x − 1.652	0.9947	59.99 ± 10.24 ^a^	48.54 ± 2.93 ^b^	42.29 ± 3.52 ^b^	41.79 ± 3.57 ^b^	34.93 ± 1.47 ^c^	23.75 ± 9.11 ^d^	16.83 ± 0.42 ^e^
3	Ser	105.1	y = 129.82x − 4.001	0.9736	26.07 ± 6.31 ^a^	27.37 ± 4.53 ^a^	24.40 ± 4.17 ^a^	21.02 ± 6.07 ^a^	14.13 ± 2.86 ^b^	14.25 ± 3.13 ^b^	10.20 ± 2.14 ^b^
4	His	155.2	y = 244.16x + 2.171	0.9947	21.11 ± 3.18 ^b^	23.06 ± 1.12 ^b^	28.31 ± 13.21 ^a^	23.94 ± 1.59 ^b^	14.41 ± 11.12 ^c^	22.66 ± 12.71 ^b^	12.17 ± 3.18 ^c^
5	Gly	75.1	y = 173.12x + 0.806	0.9867	2.64 ± 0.25 ^a^	1.94 ± 0.51 ^b^	1.01 ± 0.08 ^b^	-	-	-	-
6	Thr	119.1	y = 118.05x − 0.813	0.9946	10.18 ± 2.13 ^a^	12.68 ± 1.09 ^a^	12.20 ± 4.36 ^a^	6.10 ± 0.04 ^b^	5.29 ± 1.52 ^b^	6.84 ± 0.57 ^b^	3.24 ± 0.04 ^c^
7	Arg	174.2	y = 272.84x − 3.179	0.9892	8.82 ± 2.98 ^b^	13.90 ± 4.38 ^a^	14.41 ± 6.38 ^a^	9.30 ± 1.73 ^b^	10.10 ± 4.79 ^b^	5.48 ± 1.98 ^c^	-
8	Ala	89.1	y = 150.76x + 3.308	0.9878	11.99 ± 3.82 ^a^	12.67 ± 4.43 ^a^	10.90 ± 2.16 ^a^	7.36 ± 1.71 ^b^	3.21 ± 1.36 ^c^	4.77 ± 1.07 ^c^	3.12 ± 0.78 ^c^
9	Tyr	181.2	y = 269.11x − 1.592	0.9831	-	-	-	-	-	-	-
10	Cys	121.2	y = 193.99x + 0.254	0.9885	13.72 ± 1.77 ^a^	15.75 ± 4.11 ^a^	10.61 ± 4.36 ^b^	9.18 ± 365.49 ^b^	3.40 ± 0.44 ^c^	2.47 ± 1.31 ^c^	1.20 ± 0.09 ^c^
11	Val	117.2	y = 111.16x − 1.667	0.9972	7.02 ± 3.53 ^b^	12.78 ± 1.03 ^a^	10.96 ± 2.35 ^a^	8.13 ± 2.74 ^b^	5.44 ± 4.43 ^c^	2.54 ± 0.82 ^d^	1.43 ± 0.62 ^d^
12	Met	149.2	y = 198.02x − 0.456	0.9906	3.13 ± 0.97 ^a^	2.65 ± 1.11 ^b^	5.00 ± 1.88 ^a^	3.89 ± 0.13 ^a^	1.21 ± 0.40 ^b^	0.23 ± 0.07 ^c^	0.58 ± 0.01 ^c^
13	Phe	165.2	y = 168.98x + 4.169	0.9953	14.50 ± 5.03 ^a^	15.39 ± 1.53 ^a^	11.21 ± 7.54 ^b^	5.96 ± 3.08 ^c^	6.16 ± 4.17 ^c^	2.66 ± 0.44 ^d^	1.75 ± 0.22 ^d^
14	Ile	131.2	y = 196.92x − 1.296	0.9882	5.13 ± 0.89 ^a^	5.80 ± 1.98 ^a^	3.78 ± 0.33 ^b^	3.28 ± 0.97 ^b^	5.94 ± 0.09 ^a^	3.80 ± 1.32 ^b^	4.63 ± 1.31 ^b^
15	Leu	131.2	y = 189.35x + 1.067	0.9743	1.74 ± 0.13 ^a^	0.47 ± 0.124 ^b^	-	-	-	-	-
16	Lys	146.2	y = 153.69x − 0.885	0.9877	8.63 ± 2.43 ^a^	6.42 ± 1.88 ^b^	5.40 ± 1.92 ^b^	5.52 ± 0.25 ^b^	3.47 ± 1.129 ^c^	3.65 ± 1.69 ^c^	1.93 ± 0.13 ^d^
17	The	174.2	y = 120.98x − 17.923	0.9913	261.12 ± 11.24 ^a^	264.12 ± 21.35 ^a^	184.18 ± 11.24 ^b^	141.92 ± 4.13 ^c^	121.12 ± 11.75 ^c^	80.19 ± 7.13 ^d^	60.34 ± 11.21 ^d^

The free amino acid abbreviations are expressed as follows: Asp: aspartic acid, Glu: glutamic acid, Ser: serine, His: histidine, Gly: glycine, Thr: threonine, Arg: arginine, Ala: alanine, Tyr: tyrosine, Cys: cysteine, Val: valine, Met: methionine, Phe: phenylalanine, Ile: isoleucine, Leu: leucine and Lys: lysine, The: theanine. The average of the three results was taken and expressed as mean ± error limit, with significance indicated by the letters a, b, c, and d.

**Table 2 foods-14-02125-t002:** Isotopic distribution patterns of identified volatile compounds with ^13^C labeled and half-labeled isotopic isomers detected in CAMOLA method experiments.

No.	Compounds	m/z ^a^	Molecular Formula	System ^b^	^13^C-Labeled	Relative Distributions of Isotopologues (%) ^c^							
						M	M + 1	M + 2	M + 3	M + 4	M + 5	M + 6	M + 7
1	3-Methylbutyraldehyde	86	C_5_H_10_O	A	all	100							
					half	100							
2	Furfuryl alcohol	98	C_5_H_6_O_2_	A	all					40	60		
					half	40				20	40		
3	Hexanal	100	C_6_H_12_O	A	all				41		59		
					half	40			30		30		
4	Furfural	96	C_5_H4O_2_	A	all						100		
					half	32				32	36		
5	Benzaldehyde	106	C_7_H_6_O	A	all	100							
					half	100							
6	5-Methylfurfural	110	C_6_H_6_O_2_	A	all					50	50		
					half	100							
7	2,3-Butanedione	86	C_4_H_6_O_2_	A	all					100			
					half	33	10	10	10	37			
8	Pyrazine	80	C_4_H_4_N_2_	A	all					100			
					half	30		50		20			
9	2-Methylpyrazine	94	C_5_H_6_N_2_	A	all						100		
					half	25		25	25		25		
10	2,5-Dimethylpyrazine	108	C_6_H_8_N_2_	A	all				100				
					half	23			50			27	
11	2,3-Dimethylpyrazine	108	C_6_H_8_N_2_	A	all				100				
					half	33			46			21	
12	2-Vinylpyrazine	106	C_6_H_6_N_2_	A	all						40	30	30
					half						40	30	30
13	2-Pyrrolecarboxaldehyde	95	C_5_H_5_NO	A	all					46	54		
					half	20	26	10	30	14			
14	2-Acetylpyrrole	109	C_6_H_7_NO	A	all								100
					half	30		70					
15	2-Acetylfuran	110	C_6_H_6_O_2_	A	all				70				30
					half	50					50		30
16	2,5-dimethyl-1-propyl-1H-pyrrole	137	C_9_H_15_N	A	all				30	70			
					half	25		10	40	7	8	10	
17	N-ethylacetamide	87	C_4_H_9_NO	A	all	29		71					
					half	100							
18	Nonanoic acid	158	C_9_H_18_O_2_	A	all	100							
					half	100							

^a^ Molecular ion. ^b^ System name. (A Glucose-theanine Maillard System.). ^c^ Values were corrected by subtracting the naturally occurring percentages of ^13^C (1.10%), ^33^S (0.76%), an ^34^ S (4.20%) in M^+^ + 1 and M^+^ + 2; the loss of hydrogen observed with the molecular ion in EI-MS was also corrected in the labeled molecular ions by the ratio (M^+^ − 1)/M.

**Table 3 foods-14-02125-t003:** Isotopic distribution patterns of identified volatile compounds with D_5_ labeled and half-labeled isotopic isomers detected in CAMOLA method experiments.

No.	Compounds	m/z ^a^	Molecular Formula	D_5-_Labeled	Relative Distributions of Isotopologues (%) ^b^					
					M	M + 1	M + 2	M + 3	M + 4	M + 5
1	Furfuryl alcohol	98	C_5_H_6_O_2_	all	100					
				half	100					
2	Hexanal	100	C_6_H_12_O	all	100					
				half	100					
3	Furfural	96	C_5_H_4_O_2_	all	100					
				half	100					
4	2-Methylfuran	82	C_5_H_6_O	all	80					20
				half	90					10
5	1-Ethyl-1H-pyrrole	95	C_6_H_9_N	all						100
				half						100

^a^ Molecular ion. ^b^ Values were corrected by subtracting the naturally occurring percentages of ^13^C (1.10%), ^33^S (0.76%), and ^34^S (4.20%) in M^+^ + 1 and M^+^ + 2; the loss of hydrogen observed with the molecular ion in EI-MS was also corrected in the labeled molecular ions by the ratio (M^+^ − 1)/M.

**Table 4 foods-14-02125-t004:** Fitting equations for key aroma-active compounds at different combinations of temperature and time.

No.	Compounds	CAS	Reaction Temperature (°C)	Fitted Equation	R^2^
1	Benzaldehyde	100-52-7	100	y = 0.2379x − 0.00247	0.9149
			110	y = 0.41551e^0.01314x^ + 0.03794	0.9148
			120	y = 0.13824e^0.01326x^ + 0.31244	0.9384
2	2-Methylpyrazine	109-08-0	100	y = 0.06502x − 4.52855	0.8195
			110	y = 0.08362x − 4.14681	0.9526
			120	y = 0.13029x − 0.59493	0.8984
3	2,6-Dimethylpyrazine	108-50-9	100	y = 0.41071x − 0.25	0.995
			110	y = 0.49107x + 0.41071	0.9938
			120	y = 2.83929x − 2.86429	0.9984
4	2-Methylfuran	534-22-5	100	y = 0.00616x − 0.31015	0.915
			110	y = 0.12865e^0.01288x^ − 0.14638	0.9018
			120	y = 0.61399e^0.0133x^ − 0.20183	0.9584

## Data Availability

The original contributions presented in this study are included in the article, further inquiries can be directed to the corresponding author.

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
