# Peer review of "Formation Mechanisms and Kinetic Modeling of Key Aroma Compounds During Qidan Tea Roasting"

_foods, 2025, doi:10.3390/foods14122125_

Round 1
Reviewer 1 Report
Comments and Suggestions for Authors
Title: In the title, the word tea must be included. Many people in the word does not what is Qidan.
I suggest this title:
Study on the formation mechanism and generation law of the main aromas during the roasting process of Qidan tea
Keywords.
The words Qidan, roasting time and aroma are in the title, please include keywords that are not in the title.
For example, furan compounds could be part of the Keywords.
Introduction
L96. Please indicate what is OAV calculations
Given the current state of knowledge and the existing research gaps, the main objectives of this study are as follows
This is confused because below you write another objective.
I suggest: Given the current state of knowledge and the existing research gaps, it is important: (1) To determine the skeletal……..
Material and Methods
L110. Qidan tea is produced
L147 It is absolutely essential to assess the overall flavor profile of Qidan before analyzing the flavor components of pre- and post-roasted Qidan samples.
It is better: The overall flavor profile of Qidan tea was assessed before analyzing the flavor components of pre- and post-roasted samples.
L 187 After extraction, insert the fiber into the inlet of the gas chromatography instrument and desorb it at 230 °C for 5 min [16].
It is better: After extraction, the fiber was inserted into the inlet of the gas chromatography instrument and desorb it at 230 °C for 5 min [16].
Results and Discusion
Please improve Table 1. It is not clear what does mean the last row 24, 35 , 24 ……
The title of Table 1 must include more information. The time is important, maybe include the words roasting time in the title.
L401 In conclusion…… This is confused because the paragraph of conclusions is below.
The paper does not include Tables 2,3 and 4 that are mentioned in the text.
Conclusion
L486. I think is not necessary to use the word conclusion in this paragraph.
Author Response
It is attached here, please have a check. If more are needed, please let me know.

Reviewer 2 Report
Comments and Suggestions for Authors
I am writing to submit my review of the manuscript entitled “Study on the formation mechanism and generation law of the main aromas during the roasting process of Qidan" for your consideration. The manuscript addresses an important and timely topic in food science, specifically the dynamic changes in aroma-active compounds during the roasting of Qidan tea, a prized Wuyi Rock Tea. The combination of sensory analysis, chemical profiling, and kinetic modeling is commendable and has the potential to offer valuable insights for both academic research and the tea industry. However, the manuscript requires major revisions to improve clarity, scientific rigor, and completeness. Below, I provide detailed comments and suggestions for improvement before it can be considered for publication.
Major Comments
Title & Abstract
Title Clarity: The title is informative but could be more concise. Consider: “Formation Mechanisms and Kinetic Modeling of Key Aroma Compounds During Qidan Tea Roasting.”
Abstract Structure: The abstract is overly dense and lacks clear structure. Please explicitly state objectives, methods, key findings, and implications in separate sentences, then a collective conclusion.
Quantitative Results: The abstract should include more specific quantitative results (e.g., % changes in aroma compounds, kinetic parameters).
Novelty Statement: Clearly highlight what is novel about your approach compared to previous studies in the beginning of the abstract.
Introduction
Literature Context: The introduction provides good background but lacks a critical review of the most recent literature (2024-2025) on tea aroma formation and kinetic modeling, it seems that this is an old written work.
Research Gaps: The gap in knowledge is mentioned but not sharply defined. Please specify what is unknown about Qidan tea aroma formation under roasting compared to other oolong teas.
Objectives: The objectives are somewhat buried. Please end the introduction with a concise, bullet-pointed list of the research aim, as well as an important implication for the industry.
Materials and Methods
Sample Description: Provide more detail on the Qidan tea samples—harvest season, leaf grade, and any pre-roasting treatments.
Roasting Conditions: Clarify whether all samples were roasted in parallel or sequentially. Was the humidity controlled?
Replicates: State the number of biological and technical replicates for each experiment.
Sensory Panel: Provide more information on panelist training and selection criteria. How was panel performance validated?
Sensory Protocol: Was palate cleansing performed between samples? What was the serving temperature?
SPME-GC-MS Parameters: The SPME protocol is incomplete. Specify fiber type, extraction time, temperature, and GC-MS conditions.
Data Analysis: Clearly describe all statistical analyses, including software used and significance thresholds.
Kinetic Modeling: Provide the equations used, fitting procedures, and criteria for model selection (e.g., R², AIC).
Results and Discussion
Data Presentation: Figures and tables are referenced but not included in the excerpt. Ensure all are clear, well-labeled, and interpretable.
Sensory Results: Present sensory data with error bars and statistical comparisons (e.g., ANOVA, post hoc tests).
Compound Identification: How were aroma-active compounds identified (e.g., retention index, standards, MS spectra)?
OAV Calculation: Describe the calculation of odor activity values (OAVs) and their thresholds.
Kinetic Data: Provide kinetic parameters (rate constants, activation energies) with confidence intervals.
Correlation Analysis: Include correlation matrices between sensory attributes and chemical data.
Depth of Discussion: The discussion is descriptive but lacks critical analysis. Compare your findings with those of recent studies on Wuyi Rock Tea and other oolongs.
Mechanistic Insights: Discuss possible biochemical pathways for key aroma compounds, referencing the Maillard reaction literature.
Limitations: Explicitly discuss limitations of your study (e.g., single temperature, lack of volatile precursors analysis).
Industrial Implications: Expand on how your findings can inform tea processing practices.
Future Directions: Suggest specific follow-up experiments (e.g., varying humidity, multi-temperature kinetics).
Conclusion
Summary: The conclusion should succinctly summarize the main findings and their significance.
Practical Recommendations: Offer actionable recommendations for tea processors based on your data.
Comments on the Quality of English LanguageLanguage: The manuscript contains several grammatical and typographical errors. Please have the text thoroughly edited by a native English speaker or professional service.
Author Response
It is attached here. If more are needed, please let me know.

Round 2
Reviewer 1 Report
Comments and Suggestions for Authors
It is a very good paper !!
Author Response
Thank you.
Reviewer 2 Report
Comments and Suggestions for Authors
Thank you for considering my review of the manuscript. Specifically, I suggest that the authors improve the presentation of the figures and tables, as they are currently not well-formatted. Additionally, the references section appears to be missing some necessary data and should be updated accordingly.
Author Response
The response is attached below, please have a check.
